# Harmonious convergence for confidence estimation in monocular depth estimation and completion

## Abstract

Confidence estimation for monocular depth estimation and completion is important for their deployment in real-world applications. Recent models for confidence estimation in these regression tasks mainly rely on the statistical characteristics of training and test data, while ignoring the information from the model training. We propose a harmonious convergence estimation approach for confidence estimation in the regression tasks, taking training consistency into consideration. Specifically, we propose an intra-batch convergence estimation algorithm with two sub-iterations to compute the training consistency for confidence estimation. A harmonious convergence loss is newly designed to encourage the consistency between confidence measure and depth prediction. Our experimental results on the NYU2 and KITTI datasets show improvements ranging from 10.91% to 43.90% across different settings in monocular depth estimation, and from 27.91% to 45.24% in depth completion, measured by Pearson correlation coefficients, justifying the effectiveness of the proposed method. We will release all the codes upon the publication of our paper.

## 1 Introduction

Monocular depth estimation and completion are fundamental tasks in 3D vision, with applications spanning autonomous driving (Hu et al., 2023), 3D scene reconstruction and completion (Nunes et al., 2024), and simultaneous localization and mapping (Tateno et al., 2017; Matsuki et al., 2024). These tasks are regarded as dense regression problems as continuous depth values are expected for dense pixels in the input images. Confidence estimation is crucial for effectively deploying these regression tasks, ensuring reliable depth predictions in real-world applications.

Numerous methods have been proposed for confidence estimation that can be applied or adapted for monocular depth estimation and completion. For instance, Upadhyay et al. (2022) proposed to leverage a Bayesian autoencoder for uncertainty estimation, approximating the underlying distribution for the outputs from the frozen neural network. Zhu et al. (2022) and Shao et al. (2023a) proposed to utilize an auxiliary branch to predict the uncertainty map through joint training. Evidential learning (Amini et al., 2020; Lou et al., 2023) has been also explored for regression tasks. However, these methods often neglect to incorporate information from the model training into the confidence estimation. Recent advances, such as training consistency (Li et al., 2023) and correctness consistency (Moon et al., 2020) show promise in mitigating overconfidence in classification tasks by leveraging training information through additional regularization. Nevertheless, these methods, designed for classification tasks with discrete outputs, are not optimized for monocular depth estimation and completion models that produce continuous value outputs.

Extending training consistency from classification problems to dense regression tasks is not trivial. One challenge is addressing spatial misalignment due to random data augmentations commonly used during training. In classification tasks, random data augmentation does not impact the image-level classification results. However, dense regression tasks require pixel-level predictions, which depend on precise spatial alignment. The second challenge is the method of calculating consistency. In previous classification tasks, consistency was determined by checking whether predictions matched subsequent predictions (Li et al., 2023) or the ground truth (Moon et al., 2020). However, this ap-

proach is unsuitable for regression tasks, where depth predictions are continuous values and cannot be guaranteed to be exactly equal.

To overcome these challenges, we propose a harmonious convergence estimation algorithm for confidence estimation in monocular depth estimation and completion. First, we introduce an intra-batch convergence estimation algorithm to erase the misalignment of training samples by random augmentations. In particular, we feed the same input data into model twice, which performs two sub-iterations in each iteration for each batch of training data. It inherently ensures that the spatial alignment of the same sample is maintained because we perform two optimizations using the same input. The convergence estimation within each batch is adopted as the training information, eliminating the need to store the intermediate models/results during the entire training process and reducing demands on memory. Inspired by the fact that confidence estimation relies on depth estimation during training, a harmonious convergence loss is newly designed to encourage consistency between the convergence of depth predictions and that of the corresponding confidence estimates.

We have conducted experiments to evaluate its effectiveness on both monocular depth estimation and completion tasks. On the NYU2 and KITTI datasets, our method achieves improvements ranging from 10.91% to 43.90% across different settings in monocular depth estimation, and from 27.91% to 45.24% in depth completion, measured by Pearson correlation coefficients. The improvements show that our proposed harmonious convergence estimation algorithm outperforms existing confidence estimation methods. The contributions of our method are summarized as follows.

- We propose a harmonious convergence estimation algorithm that integrates training consistency into confidence estimation for monocular depth estimation and completion tasks.
- The proposed method adopts a novel intra-batch convergence estimation algorithm for consistency computation to overcome the challenges in computing training consistency for monocular depth estimation and completion tasks.
- We design a novel harmonious convergence loss to align the convergence of confidence estimation with that of depth prediction.
- We validate our approach through comprehensive experiments on monocular depth estimation and completion tasks. The results show the effectiveness of the proposed algorithms.

## 2 RELATED WORK

### 2.1 MONOCULAR DEPTH ESTIMATION AND COMPLETION

**Monocular depth estimation** is a fundamental application in 3D vision. The pioneering neural networks for monocular depth estimation are designed to leverage both local and global features (Eigen et al., 2014) or as a fully convolutional architecture (Laina et al., 2016). Subsequent approaches have explored various strategies to enhance monocular depth estimation performance, such as multi-scale features aggregation (Lee et al., 2019; Aich et al., 2021; Huynh et al., 2020; Lee et al., 2021), neural conditional random fields (Yuan et al., 2022), geometric constraints (Shao et al., 2024a; 2023b; Patil et al., 2022; Bae et al., 2022). For example, Bae et al. (2022) leverage surface normal and its uncertainty to recurrently refine the predicted depth-map. Then, Ranftl et al. (2021) proposed to use vision transformers (Dosovitskiy et al., 2020) instead of convolutional backbones, leveraging a global receptive field in the encoder. Built on this method, transformer-based approaches (Bhat et al., 2023) have set a new milestone for monocular depth estimation, benefiting from extensive labeled and unlabeled training data. Recently, foundational models, such as Depth Anything (Yang et al., 2024a) and Depth Anything v2 (Yang et al., 2024b), have been introduced for robust monocular depth estimation. We choose two recent and representative works, NewCRFs (Yuan et al., 2022) and Depth Anything (Yang et al., 2024a), as our main algorithms to evaluate the proposed confidence estimation algorithms for monocular depth estimation.

**Depth completion** has also attracted increasing attentions, leading to the emergence of numerous approaches in recent years. Unlike monocular depth estimation, depth completion methods introduce irregularly distributed, extremely sparse data obtained from LiDAR or structure from motion. Many approaches have been proposed to address the challenges in depth completion via multi-modal fusion, including early-fusion (Ma & Karaman, 2018; Imran et al., 2019; Ma et al., 2019), and late-fusion scheme (Tang et al., 2020; Yan et al., 2022; Yang et al., 2019). Geometry information, like

surface normal, is often introduced as intermediate representation for fusion (Chen et al., 2019; Zhao et al., 2021; Shao et al., 2024a). Depth refinement methods (Cheng et al., 2020; Park et al., 2020; Lin et al., 2022; Liu et al., 2022) mostly follow the spatial propagation mechanism (Liu et al., 2017), which iteratively refines the regressed depth by a local linear model with learned affinity. We choose two recent representative works, CompletionFormer (Zhang et al., 2023) and BPnet (Tang et al., 2024), to evaluate our proposed method for depth completion task.

## 2.2 CONFIDENCE ESTIMATION

Bayesian-based methods are often used for confidence or uncertainty estimation. These approaches treat model parameters as distributions rather than fixed values, which capture epistemic (Blundell et al., 2015; Daxberger et al., 2021; Welling & Teh, 2011; Gal & Ghahramani, 2016) and aleatoric (Kendall & Gal, 2017; Bae et al., 2021; Qu et al., 2021) uncertainties. These approaches with from-scratch training need inevitable computational expense of optimization with a large number of parameters. Monte Carlo dropout (Gal & Ghahramani, 2016) is a well-known approach that treats dropout as Bernoulli-distributed random variables, approximating the training process through variational inference. Deterministic neural network offers a more efficient estimation approach by directly computing the uncertainty of prediction distributions with a single forward pass. Deep evidential regression (Amini et al., 2020) extends the approach in classification (Sensoy et al., 2018) to regression tasks by estimating the parameters of a normal inverse gamma distribution over an underlying normal distribution, enabling explicit representation of both epistemic and aleatoric uncertainties. To address performance degradation caused by "zero confidence regions" (Pandey & Yu, 2023), Ye et al. (2024) introduced a novel uncertainty regularization term that allows the model to bypass high-uncertainty areas and effectively learn from the low-confidence regions. Recently, Xiang et al. (2024) proposed to model the uncertainty of MDE models from the perspective of the inherent probability distributions originating from the depth probability by introducing additional training regularization terms. For non-probabilistic neural networks-based methods, the log-likelihood maximization method is trained to simultaneously optimize both the original regression task and uncertainty predictions (Kuleshov et al., 2018; Song et al., 2019; Zelikman et al., 2020). Deep ensemble approaches (Lakshminarayanan et al., 2017; Wen et al., 2020) combine predictions from multiple models with varying architectures and have become increasingly popular for uncertainty modeling in recent years. Mi et al. (2022) proposed augmenting inputs with tolerable perturbations, which are then fed into a pre-trained depth estimation model to obtain different depth predictions. The differences between these outputs are used as a surrogate for uncertainty estimation. Although significant progress has been achieved, these methods fail to take the information from training process into consideration.

Recent advances using training consistency as a regularization show promising performances in confidence estimation for classification. Moon et al. (2020) proposed the correctness consistency, the frequency of correct predictions through the training process, to approximate the confidence of a model on each training sample. Li et al. (2023) then defined a prediction consistency. Given a sample $x$, the prediction consistency is defined as the frequency of a training datum getting the same prediction in sequential training epochs:

$$c = \frac{1}{M-1} \sum_{m=1}^{M-1} \mathbb{1} \left\{ \hat{y}^m = \hat{y}^{m+1} \right\} \tag{1}$$

where $\hat{y}^m$ means the prediction of sample $x$ at the $m$-th epoch, $M$ denotes the number of epochs in training. However, these methods are proposed for classification tasks and are not applicable to regression tasks. We propose a harmonious convergence estimation to extend training consistency to the depth estimation and completion, which are regression tasks.

## 3 METHODOLOGY

### 3.1 MOTIVATION

As shown in Eq. (1), the training consistency in classification can be computed by comparing the classification label and ground truth label directly. An intuitive idea to adopt this for regression tasks is to apply Eq. (1) directly. Given an image $\mathcal{X}$, the training consistency in regression is defined as the

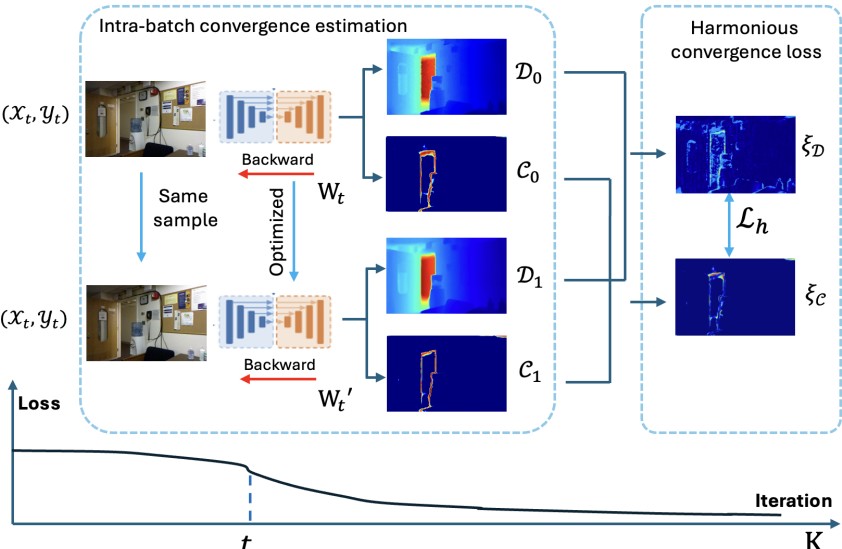

Figure 1: The overall architecture of our proposed harmonious convergence for confidence estimation. The intra-batch convergence estimation performs two forward-backwards operations in each iteration. Given a batch of training data, we first obtain the depth prediction $\mathcal{D}_0$ and its corresponding confidence $\mathcal{C}_0$. Subsequently, the same batch of training data is fed into the updated model, producing the second depth prediction $\mathcal{D}_1$ and confidence $\mathcal{C}_1$. Then, we can achieve the depth prediction convergence $\xi_\mathcal{D}$ and confidence convergence $\xi_\mathcal{C}$. A harmonious convergence loss is proposed to introduce the training convergence information into model training.

frequency with which each pixel's prediction remains consistent across sequential training epochs, as follows:

$$c = \frac{1}{M-1} \sum_{t=1}^{M-1} \sum_{i=0}^{H-1} \sum_{j=0}^{W-1} \mathbb{1}\left\{ \hat{y}_{i,j}^m = \hat{y}_{i,j}^{m+1} \right\}, \tag{2}$$

where $\hat{y}_{i,j}^t$ means the predicted outcome at position $(i,j)$ of sample $x$ at the $m$-th epoch, and $H, W$ represent the height and width of sample $x$.

However, simply extending consistency on depth prediction, as shown in Eq. (2), faces several challenges: Firstly, monocular depth estimation and completion yield pixel-wise outputs that require spatial consistency and alignment for accurate computation of consistency. However, augmentations such as random cropping would destroy this spatial consistency. Secondly, both tasks are regression tasks predicting continuous valued outputs, different from discrete valued outputs in classifications. We would get plenty of zeros from Eq. (2). A possible way is to modify it with a threshold to reject small differences, however, this would leads to the loss of nuanced information and arbitrary decisions. To address the above challenges, we propose a harmonious convergence estimation algorithm. It includes a novel intra-batch convergence estimation algorithm which performs two sub-iterations in each iteration for each batch of training data, along with a newly designed harmonious convergence loss.

## 3.2 HARMONIOUS CONVERGENCE ESTIMATION

### 3.2.1 INTRA-BATCH CONVERGENCE ESTIMATION

Intra-batch convergence estimation performs two sub-iterations in each iteration and compute the consistency between the two sub-iterations. This is different from previous algorithms (Li et al., 2023) that compute consistency among models after different epochs of training.

As shown in Fig 1, the two sub-iterations involves the forward-backward optimization using the same batch of augmented training data. In the first step, given one batch of training samples $\mathcal{X}_t$ at

iteration $t$ and a prediction model with parameters $W_t$, we achieve the prediction result, $\mathcal{D}_0$, and the confidence, $\mathcal{C}_0$. After computing the loss, the model parameters are updated to $W_t'$ from $W_t$ with backward optimization. In the second step, we input the same batch of training samples $\mathcal{X}_t$ with the same augmentation into the model with the updated parameters $W_t'$, obtaining the second-step prediction result $\mathcal{D}_1$ and the second-step confidence map $\mathcal{C}_1$. As the same augmentation is used, we define and compute a depth prediction convergence $\xi_{\mathcal{D}}$ by directly comparing the outputs as follows.

$$\xi_{\mathcal{D}} = \frac{\|\mathcal{D}_1 - \mathcal{D}_0\|}{\mathcal{D}_0} \tag{3}$$

The depth prediction convergences is used to compute a harmonious convergence loss for model training, which explained in more details later in Section 3.2.2.

Compared with computing training consistency among models after different epochs of training, the advantages of our proposed intra-batch convergence estimation are two-fold. First, it inherently ensures that the spatial alignment of the same sample is maintained because we perform two optimizations using the same input. Second, convergence estimation is calculated within each batch. It eliminates the need to store the intermediate models/results during the entire training process, reducing demands on memory which can be significantly large for dense regression task such as monocular depth estimation and completion.

### 3.2.2 HARMONIOUS CONVERGENCE LOSS

As the main model for depth prediction converges, it is expected that the confidence of the depth prediction to stabilize as well. Motivated by that, we define a confidence convergence $\xi_{\mathcal{C}}$ for confidence estimation, which is expected to be consistent with $\xi_{\mathcal{D}}$:

$$\xi_{\mathcal{C}} = \frac{\|\mathcal{C}_1 - \mathcal{C}_0\|}{\mathcal{C}_0}. \tag{4}$$

To achieve consistence between $\xi_{\mathcal{C}}$ and $\xi_{\mathcal{D}}$, a straightforward way is to compute their absolute difference or mean square difference. However, we observe higher $\xi_{\mathcal{C}}$ than $\xi_{\mathcal{D}}$ in such a method. We analyzed the training process and realized that this discrepancy arises because the ground truth depths are available for depth prediction model training, while the confidence prediction model relies on the convergence of depth prediction models.

Motivated by the above observations, a harmonious convergence loss $\mathcal{L}_h$ is newly designed to encourage the convergence of the confidence prediction to be consistent with that of the depth prediction. Formally,

$$\mathcal{L}_h = \sum_{i=0}^{H-1} \sum_{j=0}^{W-1} \max\{0, \xi_{\mathcal{D}}(i,j) - \text{sgn}(\mathcal{D}_1 - \mathcal{D}_0)\xi_{\mathcal{C}}(i,j)\}, \tag{5}$$

where $i, j$ denotes the horizontal and vertical coordinates of the pixels and $\text{sgn}(\cdot)$ denotes the sign function. When $\mathcal{D}_1 > \mathcal{D}_0$, the confidence estimation is learned to converge similarly to that for the depth prediction through training.

### 3.3 JOINT DEPTH PREDICTION AND CONFIDENCE ESTIMATION

In our implementation, we adopt a multitask learning approach for joint depth prediction and confidence estimation. This is accomplished by adding a new branch for confidence estimation on top of the existing depth prediction network.

**Monocular Depth Estimation and Completion.** The monocular depth estimation and completion tasks aim to estimate a pixel-wise depth map or complete dense depth map from a sparse one. Given an image $\mathcal{X}$ and its corresponding depth ground truth $\mathcal{D} \in \mathbb{R}^{H \times W}$, the training objective is to learn a mapping to output depth $\hat{\mathcal{D}}$ by minimizing the depth estimation loss $\mathcal{L}_{\mathcal{D}}$.

**Confidence Estimation.** The confidence in this work is defined as the posterior probability (Kendall & Gal, 2017; Zhu et al., 2022) in monocular depth estimation and completion models. The confidence map $\mathcal{C}$ indicates the pixel-wised confidence or certainty of the predictions. It has the same size as the predicted depth map, with each value representing the model's confidence of the depth

prediction. We use a simple structure for the confidence estimation. It consists of three convolutions and a Sigmoid activation function to ensure that $\mathcal{C}$ falls within the range of (0, 1). During the training process, we hope to minimize a confidence estimation loss $\mathcal{L_C}$ as in (Zhu et al., 2022):

$$\mathcal{L_C} = \lambda \cdot \mathcal{C} \cdot (\hat{\mathcal{D}} - \mathcal{D})^2 - \log(\mathcal{C}) \tag{6}$$

where $\lambda$ is used to control the overall range of the confidence map.

## 3.4 LOSS FUNCTION

The overall loss $\mathcal{L}$ is computed by combining the depth estimation loss, the harmonious convergence loss and the confidence estimation loss as follows,

$$\mathcal{L} = \mathcal{L_D} + \mathcal{L_C} + \gamma \mathcal{L}_h, \tag{7}$$

where $\gamma$ represents the weight of harmonious convergence loss. After computing the loss $\mathcal{L}$, the second forward-backwards optimization is used to update the model parameters.

## 4 EXPERIMENT

### 4.1 EVALUATION PROTOCOL

The evaluation protocol is designed to evaluate the performance when integrating a confidence estimation method with a monocular depth estimation or completion method. With similar accuracy, a higher confidence level indicates a higher reliability of the regression model.

In this paper, we evaluate our algorithm for confidence estimation in monocular depth estimation and completion tasks. We use recent state-of-the-art methods as backbones for each task, namely, NewCRFs and Depth Anything for monocular depth estimation, CompletionFormer and BPnet for depth completion. Then, we combine the proposed confidence estimation algorithm with these depth prediction backbones and follow the training setting of backbones to retrain or finetune the models.

To estimate the confidence level, we use the following metrics: the Pearson correlation coefficient, Spearman correlation coefficient, and the Area Under the Sparsification Error (AUSE). We employ correlation metrics to evaluate the relationship between the confidence map error (1-$\mathcal{C}$) and prediction error. Specifically, we calculate Pearson and Spearman correlation coefficients to quantify this relationship in our study. As in (Ilg et al., 2018; Poggi et al., 2020; Hornauer & Belagiannis, 2022), we compute AUSE that is the difference between the sparsification and the oracle sparsification. The oracle sparsification is given if the uncertainty ranking corresponds to the ranking of the true error.

At the same time, we also report the commonly used metrics to evaluate the performance of the depth prediction tasks, such as absolute relative error (Abs.Rel), scale invariant logarithmic error (SILog) and "$\delta < 1.25$" for monocular depth estimation, root mean square error (RMSE) and mean absolute error (MAE) for depth completion. Although our main objective here is not to improve the performance of the monocular depth estimation or completion, it is important to show that including the confidence estimation would not lead to a performance drop in the original tasks.

### 4.2 MONOCULAR DEPTH ESTIMATION

#### 4.2.1 EXPERIMENTAL SETTINGS

**Monocular Depth Estimation Algorithms.** Two recent and representative works, NewCRFs (Yuan et al., 2022) and Depth Anything (Yang et al., 2024a), are employed as examples for evaluating the effectiveness of the proposed confidence estimation in monocular depth estimation. NewCRFs (Yuan et al., 2022) introduced a neural window fully connected CRFs and embedded it into the depth prediction network. We choose this algorithm as it is a representative work in recent years and it inspires many subsequent novel approaches Shao et al. (2024b;c). Depth Anything (Yang et al., 2024a) offers a highly practical solution for robust monocular depth estimation. Rather than focusing on novel technical modules, this approach establishes a simple yet powerful foundational model capable of handling any images under any circumstances. We choose this algorithm as it is one of the latest method based on foundation models.

**Confidence Estimation Baselines.** We employ BayesCap (Upadhyay et al., 2022), UR-Evidential (Ye et al., 2024), and GrUMoDepth (Hornauer & Belagiannis, 2022) as the uncertainty estimation baselines. BayesCap proposes a Bayesian identity cap for uncertainty estimation, freezing the neural network parameters without affecting the trained model's performance. GrUMoDepth is a post hoc uncertainty estimation approach for an already trained depth estimation model. The UR-Evidential algorithm introduces an uncertainty regularization term for the original evidential regression learning, improving uncertainty estimation's robustness. The key difference between ours and baselines is that our method introduces a consistency constraint during training.

**Datasets.** We use two commonly-used public datasets from indoor depth estimation to outdoor depth estimation, including NYUv2 (Silberman et al., 2012), KITTI (Geiger et al., 2012) The NYUv2 dataset comprises 120K RGB-D video frames captured from 464 indoor scenes, making it a standard benchmark for indoor environments. The KITTI dataset is a widely used benchmark featuring outdoor scenes captured from a moving vehicle. We adhered to the training/testing split used in NewCRFs (Yuan et al., 2022) to ensure a fair evaluation.

**Implementation Details.** For NewCRFs (Yuan et al., 2022), we implemented our approach alongside three confidence estimation methods and conducted evaluation experiments. All networks were optimized end-to-end using the Adam optimizer ($\beta = 0.9$). The training runs for 20 epochs with a batch size of 8 and the learning rate decreasing from $1 \times 10^{-4}$ to $1 \times 10^{-5}$. The Depth Anything (Yang et al., 2024a) is a foundation-based model trained with a large number of data. Since full training from scratch is not feasible, we load the pre-trained model weights and fine-tune the encoder of the Depth Anything model together with the branch for confidence estimation.

### 4.2.2 PERFORMANCE COMPARISON

We integrate the proposed convergence stability with NewCRFs and Depth Anything, and compare with the three uncertainty estimation baseline methods on NYUv2 and KITTI datasets.

Table 1 summarizes the performance comparison with different confidence estimation algorithms including BayesCap (Upadhyay et al., 2022), GrUmoDepth (Hornauer & Belagiannis, 2022), and UR-Evidential (Ye et al., 2024) on the NYUv2 dataset. Overall, the results across four different evaluation metrics consistently indicate that our proposed method successfully adapts the models better than other baseline approaches. In particular, our method achieves 0.63 and 0.59 of Pearson metric, respectively, on NewCRFs and Depth Anything, making a comparative improvement of 21.15% and 13.46% against the best-performing baseline. Accordingly, the AUSE decreases by 4.94% from 0.085 to 0.081 and 8.33% from 0.048 to 0.044 for NewCRFs and Depth Anything, respectively. At the same time, the performance of the monocular depth estimation is maintained or slightly improved as measured by Abs Rel.

Table 2 details the performance comparisons on the KITTI dataset. Similar to the experimental results on NYUv2, our proposed method surpasses other confidence estimation methods for both NewCRFs and Depth Anything in monocular depth estimation. The Pearson correlation coefficients improved by 10.91% and 43.90% in KITTI dataset for NewCRFs and Depth Anything respectively.

Table 1: Performance Comparison for Confidence Estimation in Monocular Depth Estimation on NYU-v2 dataset.

| Methods | Pearson ↑ | Spearman ↑ | AUSE ↓ | Abs Rel ↓ | $\delta < 1.25$ ↑ |
|---|---|---|---|---|---|
| NewCRFs | / | / | / | 0.095 | 0.922 |
| + BayesCap [ECCV22] | 0.45 | 0.52 | 0.089 | 0.094 | 0.926 |
| + GrUmoDepth [ECCV22] | 0.51 | 0.58 | 0.084 | 0.095 | 0.923 |
| + UR-Evidential [AAAI24] | 0.52 | 0.61 | 0.085 | 0.094 | 0.925 |
| Ours | **0.63** | **0.68** | **0.081** | **0.093** | **0.931** |
| Depth Anything | / | / | / | 0.053 | 0.972 |
| + BayesCap [ECCV22] | 0.44 | 0.47 | 0.049 | 0.053 | 0.971 |
| + GrUmoDepth [ECCV22] | 0.52 | 0.59 | 0.050 | 0.053 | 0.972 |
| + UR-Evidential [AAAI24] | 0.51 | 0.53 | 0.048 | 0.051 | 0.975 |
| Ours | **0.59** | **0.64** | **0.044** | **0.049** | **0.978** |

Table 2: Performance comparison for confidence estimation in monocular depth estimation on KITTI dataset.

| Methods | Pearson ↑ | Spearman ↑ | AUSE ↓ | SILog ↓ | $\delta < 1.25$ ↑ |
|---|---|---|---|---|---|
| NewCRFs | / | / | / | 8.31 | 0.968 |
| + BayesCap [ECCV22] | 0.41 | 0.49 | 6.92 | 7.78 | 0.971 |
| + GrUmoDepth [ECCV22] | 0.55 | 0.51 | 6.87 | 7.54 | 0.973 |
| + UR-Evidential [AAAI24] | 0.49 | 0.53 | 7.02 | 7.91 | 0.972 |
| Ours | **0.61** | **0.65** | **6.56** | **7.32** | **0.975** |
| Depth Anything | / | / | / | 5.88 | 0.979 |
| + BayesCap [ECCV22] | 0.5 | 0.57 | 5.63 | 5.81 | 0.979 |
| + GrUmoDepth [ECCV22] | 0.39 | 0.43 | 5.54 | 5.65 | 0.980 |
| + UR-Evidential [AAAI24] | 0.41 | 0.48 | 5.62 | 5.73 | 0.979 |
| Ours | **0.59** | **0.65** | **5.41** | **5.49** | **0.982** |

### 4.2.3 ABLATION STUDIES AND ANALYSIS

**Effectiveness of $\mathcal{L}_{\mathcal{C}}$ and $\mathcal{L}_h$.** We first investigated the effectiveness of the harmonious loss and the confidence estimation loss on monocular depth estimation. We use the network from NewCRFs for depth estimation. The original NewCRFs does not provide a confidence. A naive joint training with $\mathcal{L}_{\mathcal{C}}$ alone leads to a confidence estimation with Pearson correlation coefficient of 0.52. Further including the proposed harmonious convergence loss, we achieve 0.63, as shown in Table 3. This indicates that our proposed consistency loss can reduce the model's overconfidence.

Table 3: The ablation study for the proposed loss on monocular depth estimation on NYUv2 dataset.

| $\mathcal{L}_{\mathcal{C}}$ | $\mathcal{L}_h$ | Pearson ↑ | Spearman ↑ | AUSE ↓ | AbsRel ↓ |
|---|---|---|---|---|---|
| | | / | / | / | 0.095 |
| ✓ | | 0.52 | 0.59 | 0.087 | 0.095 |
| ✓ | ✓ | 0.63 | 0.68 | 0.081 | 0.093 |

**Effects of differernt $\lambda$.** $\lambda$ controls the range of confidence map. We have conducted experiments for three different scales at 0.01, 0.1 and 1. Table 4 presents a comparison of results for different $\lambda$ values. Our studies show that $\lambda = 0.1$ gives the optimal results and we use this value in all experiments in this paper.

Table 4: The performance comparison for different $\lambda$ in monocular depth estimation

| | Pearson ↑ | Spearman ↑ | AUSE ↓ | AbsRel ↓ |
|---|---|---|---|---|
| NewCRFs | / | / | / | 0.095 |
| $\lambda = 1$ | 0.58 | 0.64 | 0.083 | 0.093 |
| $\lambda = 0.1$ | 0.63 | 0.68 | 0.081 | 0.093 |
| $\lambda = 0.01$ | 0.55 | 0.61 | 0.087 | 0.094 |

**The weight $\gamma$ of harmonious convergence loss.** Table 5 presents a comparison of results for different $\gamma$ values. We set the weight of the harmonious convergence loss at three scales: 2, 1, and 0.5. The experiments show that the performance is optimal when $\gamma$ is set to 1. The impact of different $\gamma$ values on the final performance is not significant, further demonstrating the effectiveness of our proposed harmonious convergence loss.

### 4.3 DEPTH COMPLETION

### 4.3.1 EXPERIMENTAL SETTINGS

**Depth Completion Algorithms.** We use two latest methods, CompletionFormer (Zhang et al., 2023) and BPnet (Tang et al., 2024), as our backbone algorithms for depth completion. CompletionFormer introduces a joint convolutional attention and transformer block, which enhances the extraction of both local and global features. BPnet propagates depth at the earliest stage to avoid

Table 5: The performance comparison for different $\gamma$ in monocular depth estimation

|  | Pearson ↑ | Spearman ↑ | AUSE ↓ | AbsRel ↓ |
|---|---|---|---|---|
| NewCRFs | / | / | / | 0.095 |
| $\gamma = 2$ | 0.61 | 0.67 | 0.082 | 0.093 |
| $\gamma = 1$ | 0.63 | 0.68 | 0.081 | 0.093 |
| $\gamma = 0.5$ | 0.62 | 0.66 | 0.081 | 0.093 |

directly convolving on sparse data, achieving state-of-the-art performance on NYUv2. We choose these two representative backbones for comparison on depth completion.

**Confidence Estimation Baselines.** Similar to that in monocular depth estimation in 4.2.1, we also implement those baselines for depth completion.

**Datasets.** We take the commonly used dataset, NYUv2, for performance evaluation. The NYUv2 dataset consists of RGB and depth images captured by Microsoft Kinect in 464 indoor scenes. We follow the previous work (Zhang et al., 2023; Tang et al., 2024) to split the training/testing datasets for evaluation. The sparse input depth is generated by random sampling from the dense ground truth.

**Implement Details.** Following the baseline CompletionFormer (Zhang et al., 2023), we implement our model using AdamW as optimizer with an initial learning rate of 0.001, $\beta_1 = 0.9$, $\beta_2 = 0/999$, weight decay of 0.01. The batch size per GPU is set to 12 on the NYUv2 dataset.

### 4.3.2 PERFORMANCE COMPARISON

We integrate the proposed convergence stability with CompletionFormer and BPnet, and compare with the three state-of-the-art confidence estimation methods, BayesCap (Upadhyay et al., 2022), GrUmoDepth (Hornauer & Belagiannis, 2022), and UR-Evidential (Ye et al., 2024) .

Table 6 summarizes the performance comparison built on on the NYUv2 dataset. We achieve an relative improvement of 45.24% and 27.91% compared with the best-performing confidence estimation algorithms in Pearson correlation coefficients for CompletionFormer and BPNet backbones respectively. Overall, the experimental results across four evaluation metrics consistently indicate that our proposed method successfully adapts the models better than other confidence estimation methods while the accuracy of depth completion is maintained. Figure 2 visualizes the comparison between our proposed method and UR-Evidential. From the visualization, we can see that our proposed method increases the correlation between the depth prediction and the confidence estimation.

Table 6: The performance comparison of depth completion on NYU-v2 dataset.

| Methods | Pearson ↑ | Spearman ↑ | AUSE ↓ | RMSE ↓ | MAE ↓ |
|---|---|---|---|---|---|
| CompletionFormer | / | / | / | 0.090 | 0.035 |
| + BayesCap [ECCV22] | 0.40 | 0.47 | 0.085 | 0.091 | 0.036 |
| + GrUmoDepth [ECCV22] | 0.42 | 0.48 | 0.084 | 0.090 | 0.035 |
| + UR-Evidential [AAAI24] | 0.38 | 0.44 | 0.087 | 0.090 | 0.035 |
| Ours | **0.61** | **0.67** | **0.081** | **0.089** | **0.035** |
| BPnet | / | / | / | 0.089 | 0.034 |
| + BayesCap [ECCV22] | 0.38 | 0.42 | 0.083 | 0.089 | 0.035 |
| + GrUmoDepth [ECCV22] | 0.39 | 0.43 | 0.082 | 0.089 | 0.034 |
| + UR-Evidential [AAAI24] | 0.43 | 0.51 | 0.081 | 0.089 | 0.034 |
| Ours | **0.55** | **0.63** | **0.080** | **0.089** | **0.034** |

## 5 CONCLUSION

Confidence estimation for dense regression tasks such as monocular depth estimation and completion is a challenging task. Existing methods for confidence estimation either fail to consider information from training process or do not apply for dense regression tasks. In this paper, we propose a harmonious convergence estimation algorithm. By adopting an intra-batch convergence algorithm

with two sub-iterations, our method is able to compute the training consistency in an efficient way. Inspired by the fact that the confidence convergence relies on depth model convergence, we also propose a harmonious convergence loss to encourage the convergence of confidence estimation to be consistent with depth prediction convergence. Our experimental results have shown the effectiveness of the proposed algorithm. In future work, we would further validate our algorithm in other regression tasks.

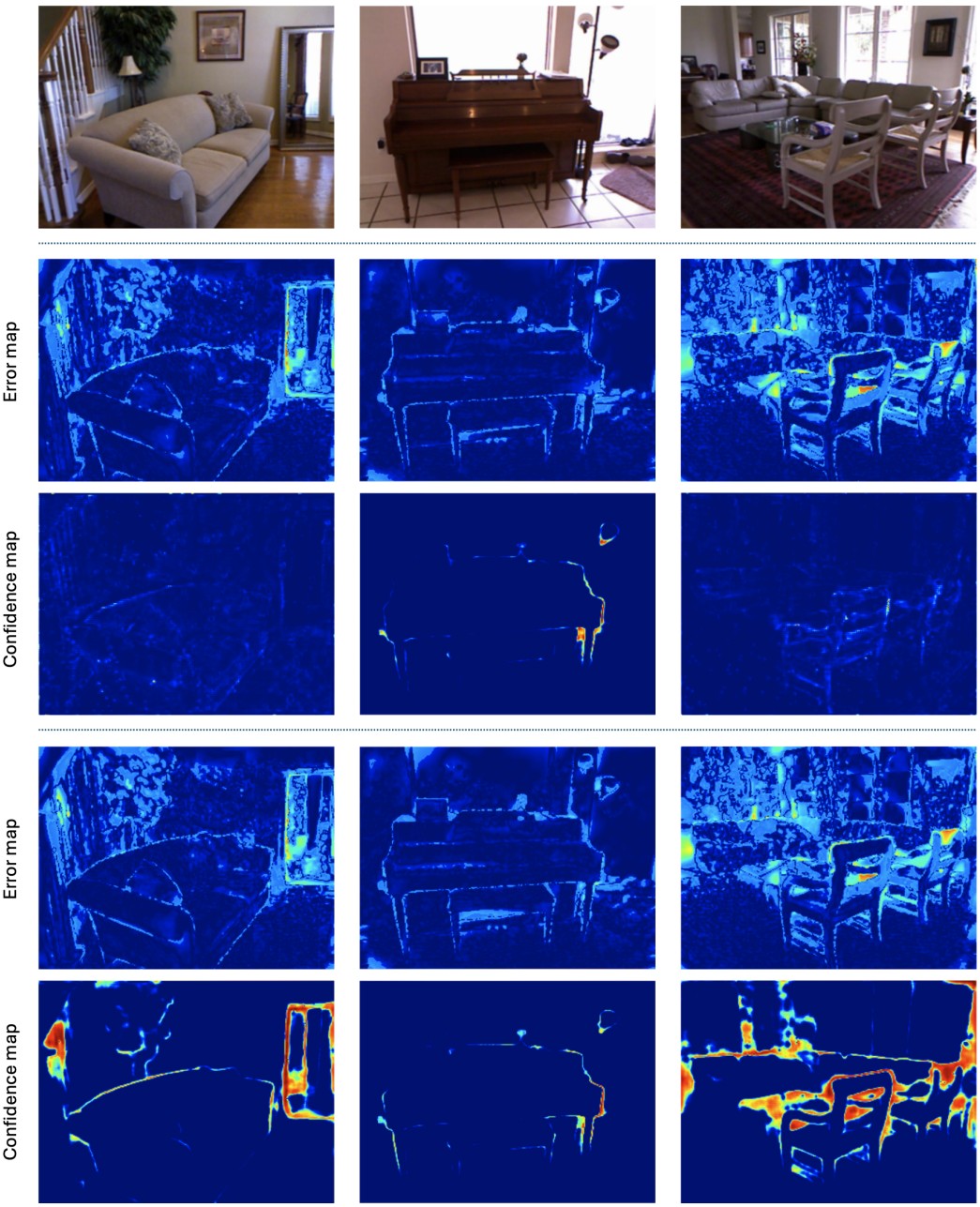

Figure 2: The visualization comparison from NYUv2 for depth completion. We choose the CompletionFormer as the backbone. The first row shows the original input images. The second and third rows show the error map and the confidence map by the previous UR-evidential. The fourth and fifth rows show the error map and confidence map of our proposed method.

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
