# OpenReview forum: "Harmonious convergence for confidence estimation in depth estimation and completion"
_ICLR.cc/2025/Conference — Submitted to ICLR 2025_

### Official Review · Reviewer_T3bM · 2024-10-30

**Soundness:** 2
**Presentation:** 1
**Contribution:** 2
**Rating:** 3
**Confidence:** 4

**Summary:**

This paper proposes a confidence estimation method in depth estimation and depth completion. The existing dense prediction confidence estimation methods may suffer high memory demands, so this paper proposes an "intra-batch convergence estimation algorithm" for consistency computation, and designs a "harmonious convergence loss function" that integrates training consistency
into confidence estimation for monocular depth estimation and completion tasks. Some experiments conducted on NYU and KITTI datasets prove the effectiveness of this method.

However, this paper fails to adequately summarize existing methods, and the proposed solution also lacks theoretical support, and some related works are missing for comparison.

**Strengths:**

1. This paper proposes an "intra-batch convergence estimation algorithm" for consistency computation, and designs a "harmonious convergence loss function" that integrates training consistency into confidence estimation for monocular depth estimation and completion tasks.
2. Some experiments conducted on NYU and KITTI datasets prove the effectiveness of this method.

**Weaknesses:**

1. This paper does not provide a good summary of the problems of existing methods. So, I can not figure out the motivation behind the entire design. As mentioned in Line 48 "One challenge is addressing spatial misalignment of training samples caused by random augmentations", but why I should overcome this point? In other words, I do not understand why we have to introduce "training consistency" into confidence estimation networks. I can not see any advantages of doing so over existing methods. The only explanation is in line 130 "Although significant progress has been achieved, these methods fail to take the information from training process into consideration." Through these words, I can not understand the author's motivation.
2. There are some missing related works for comparison (Bae G et al. 2022, Xiang M et al. 2024). But the most important is I need to know the Strengths between this paper and these existing determined confidence estimation methods.

3. The proposed solution in the papers lacks theoretical support. In my opinion, the inter-batch loss is just a constraint that ensures the gradient is sufficiently small during the optimization process of the network. This is a simple modification of (Li et al. 2023) without corresponding explanations. And the ablation experiment (Table 3) also shows that the $L_h$  is the most important part of the whole network, not $L_c$.

[1]  Bae G, Budvytis I, Cipolla R. Irondepth: Iterative refinement of single-view depth using surface normal and its uncertainty[J]. arXiv preprint arXiv:2210.03676, 2022.

[2]  Xiang M, Zhang J, Barnes N, et al. Measuring and Modeling Uncertainty Degree for Monocular Depth Estimation[J]. IEEE Transactions on Circuits and Systems for Video Technology, 2024.

[3] Chen Li, Xiaoling Hu, and Chao Chen. Confidence estimation using unlabeled data. In The Eleventh International Conference on Learning Representations (ICLR), 2023.

**Questions:**

See in the Weaknesses part.

---

> ### Author Response · Authors · 2024-11-25
> **Reply to reviewer T3bM**
>
> We sincerely thank the reviewer for their detailed feedback. We appreciate the acknowledgement of our thorough experiments on the NYU and KITTI datasets.
>
> ## 1. Response to the weakness of our summary is not clear
> Thank you to the reviewer for pointing out the issue. We have revised and reorganized some descriptions in the introduction to articulate our motivation better.
>
> **Overall statements**:
>
> Our approach makes a significant contribution by incorporating training information into confidence estimation for depth estimation and completion tasks to mitigate the issue of model overconfidence.
>
> **Motivation**:
>
> Previous evidential learning-based confidence estimation approaches typically focus on the statistical characteristics of the training and testing datasets for depth estimation and completion tasks without leveraging the valuable insights available from the model's training process. As described in Lines 43-45, Recent advances, such as training consistency (Li et al., 2023) [1] and prediction consistency (Moon et al., 2020) [2], show promise in mitigating overconfidence in classification tasks by leveraging training information through additional regularization. Therefore, our motivation is to consider training information for depth estimation and completion tasks.
>
> **Problem Statements**:
>
> Recent methods have demonstrated the effectiveness of using training information (including prediction consistency and correctness consistency) to reduce overconfidence in classification tasks. These methods leverage training information, defined as the predictions of all samples throughout the training process. However, extending this concept from classification to dense regression tasks such as depth estimation and completion presents two key challenges:
>
> 1) Spatial Misalignment: The first challenge arises from spatial misalignment caused by random data augmentations commonly used during training. While such augmentations do not affect image-level classification results in classification tasks, dense regression tasks require pixel-level predictions that rely heavily on precise spatial alignment.
>
> 2) Consistency Calculation: The second challenge lies in calculating consistency. In previous classification tasks, consistency was determined by evaluating whether predictions matched subsequent predictions [1] or the ground truth [2]. However, this approach does not apply to regression tasks, where depth predictions are continuous values and cannot be expected to match exactly. An alternative approach involves translating consistency into a threshold-based comparison, but identifying an appropriate and effective threshold remains a significant challenge.
>
> **Our contribution**:
>
> 1) We propose a harmonious convergence estimation algorithm incorporating training consistency into confidence estimation for monocular depth estimation and completion tasks.
>
> 2) The proposed method introduces a novel intra-batch convergence estimation algorithm for consistency computation, effectively addressing the challenges of calculating training consistency in monocular depth estimation and completion tasks.
>
> 3) We design a harmonious convergence loss to align the convergence of confidence estimation with the convergence of depth prediction.

---

> > ### Author Response · Authors · 2024-11-25
> > **Supplemental reply to reviewer T3bM**
> >
> > ## 2. Response to missing some related works.
> > We sincerely appreciate the reviewer for pointing out the missing related works.
> >
> > IronDepth [3] leverages surface normal and its uncertainty to iteratively refine the predicted depth map. However, our work does not aim to improve the accuracy of depth prediction itself but rather to enhance the confidence estimation for depth prediction. Xiang et al. [4] propose additional training regularization between depth estimation and confidence estimation. This approach still relies on statistical characteristics without incorporating training information into the training process, which is different from our contribution.
> >
> > We have included these works in the revised version of the manuscript.
> >
> > ## 3. Response to the lack of theoretical support
> >
> > Training consistency is proposed in [1] and demonstrates its effectiveness in classification tasks. Our paper aims to extend training consistency to dense regression tasks, specifically for depth estimation and completion. This extension faces significant challenges posed by regression tasks, such as the definition of consistency in regression and spatial misalignments.
> >
> > Our primary contribution lies in bridging the gap for introducing training consistency into regression tasks, which requires novel designs, such as intra-batch convergence estimation and harmonious convergence loss. Experimental results verify the effectiveness of our proposed method.
> >
> > ## 4. Response to the ablation study
> >
> > Regarding the ablation experiment in Table 3, we appreciate the reviewer's observation and would like to clarify the results:
> >
> > 1)  $L_c$: This is the confidence estimation loss, which incorporates the confidence estimation into existing depth estimation and completion approaches.
> > 2) $L_h$: This is the proposed Harmonious Convergence Loss, which integrates training consistency into the model.
> >
> > The results in Table 3 show that $L_h$ is the most critical component of the network, as it captures and enforces training consistency, driving performance improvements. $L_c$ is essential for implementing confidence estimation for existing depth estimation and completion approaches.
> >
> > [1] Chen Li, Xiaoling Hu, and Chao Chen. Confidence estimation using unlabeled data. In The Eleventh International Conference on Learning Representations (ICLR), 2023.
> >
> > [2] Moon J, Kim J, Shin Y, et al. Confidence-aware learning for deep neural networks[C]//international conference on machine learning. PMLR, 2020: 7034-7044.
> >
> > [3] Bae G, Budvytis I, Cipolla R. Irondepth: Iterative refinement of single-view depth using surface normal and its uncertainty[J]. arXiv preprint arXiv:2210.03676, 2022.
> >
> > [4] Xiang M, Zhang J, Barnes N, et al. Measuring and Modeling Uncertainty Degree for Monocular Depth Estimation[J]. IEEE Transactions on Circuits and Systems for Video Technology, 2024.

---

> ### Author Response · Authors · 2024-11-27
>
> Dear reviewer T3bM,
>
> Given that the rebuttal phase is about to end, we would like to know if our responses have addressed your concerns. Additionally, we are still here and happy to address any further concerns.

---

### Official Review · Reviewer_had4 · 2024-11-01

**Soundness:** 3
**Presentation:** 3
**Contribution:** 2
**Rating:** 5
**Confidence:** 2

**Summary:**

This paper addresses the confidence estimation for monocular depth estimation and completion. To this end, the authors propose a intra-batch convergence estimation algorithm and a harmonious convergence loss. The experiments are conducted on the NYU2 and KITTI datasets and the results show improvements ranging from 10.91% to 43.90% across different settings in monocular depth estimation, and from 27.91% to 45.24% in depth completion.

**Strengths:**

The paper is well-written and structured;

The experiments are thorough and conducted on multiple datasets;

The proposed method is effective according to the quantitative and qualitative results.

**Weaknesses:**

Could you provide a more detailed analysis of how intra-batch convergence estimation or harmonious convergence loss contribute to the performance gains compared to previous approaches?

In Fig. 1, D0 and D1 appear to be different, but C0 and C1 look exactly the same；Could you explain this apparent discrepancy and discuss its implications for the method's effectiveness? Additionally, could you provide a more detailed visualization or analysis of how C0 and C1 differ?

**Questions:**

Please see the weakness.

---

> ### Author Response · Authors · 2024-11-25
> **Reply to reviewer had4**
>
> We sincerely thank the reviewer had4 for thoughtful feedback and recognition of the strengths of our paper, including the clear writing, structured presentation, thorough experiments, and the effectiveness of the proposed method.
>
> ## 1. Response to detailed analysis of our approaches.
> ### 1.1Problem statements:
> Recent methods have demonstrated the effectiveness of leveraging training information (including prediction consistency [1] and correctness consistency [2]) to mitigate overconfidence in classification tasks. However, extending this concept to dense regression tasks such as depth estimation and completion faces the following key challenges:
>
> 1) The first challenge is the spatial misalignment due to random data augmentations commonly used during training. In classification tasks, random data augmentation does not impact the image-level classification results. However, dense regression tasks require pixel-level predictions, which depend on precise spatial alignment.
>
> 2) The second challenge is the method of calculating consistency. In previous classification tasks, consistency was determined by checking whether predictions matched subsequent predictions [1] or the ground truth [2]. However, this approach is unsuitable for regression tasks, where depth predictions are continuous values and cannot be guaranteed to be exactly equal. An alternative solution is translating training consistency into a comparison with a threshold; however, determining an appropriate and effective threshold value poses a significant challenge.
>
> ### 1.2 Contribution Elaboration of Intra-Batch Convergence Estimation and Harmonious Convergence Loss:
> To address these challenges, we propose intra-batch convergence estimation, an efficient method to capture training information specifically for depth estimation and completion tasks. The key idea is to infer the model on the same sample twice within a mini-batch and compute the convergence between the two predictions as the training information. This approach avoids the memory bottlenecks of tracking predictions of all samples. Additionally, comparing predictions from the same augmented input naturally accommodates spatial misalignment. As shown in Table 1, based on NewCRFs, our algorithm achieves significant improvements compared to the latest evidential learning-based method, with a 21.2% increase in Pearson correlation (from 0.52 to 0.63) and an 11.1% increase in Spearman correlation (from 0.61 to 0.68).
>
> We also design the harmonious convergence loss, integrating convergence information into the model training. Together, these two enable us to introduce training information into the training of regression tasks, addressing both challenges. The result is improved confidence estimation, as demonstrated by the performance gains in Table 3.
>
> ## 2. Clarification Regarding Fig. 1.
> We deeply appreciate the reviewer pointing out the issue in Fig. 1, where C0 and C1 appear identical. This is an oversight in the visualization process. We have corrected this error in the revised version of the paper and updated the figure to reflect the process accurately. We believe these revisions address the concern and enhance the clarity of our work.
>
> [1] Chen Li, Xiaoling Hu, and Chao Chen. Confidence estimation using unlabeled data. In The Eleventh International Conference on Learning Representations (ICLR), 2023.
>
> [2] Moon J, Kim J, Shin Y, et al. Confidence-aware learning for deep neural networks[C]//international conference on machine learning. PMLR, 2020: 7034-7044.

---

> ### Author Response · Authors · 2024-11-27
>
> Dear Reviewer had4,
>
> Given that the rebuttal phase is about to end, we would like to know if our responses have addressed your concerns. Additionally, we are still here and happy to address any further concerns.

---

### Official Review · Reviewer_LDwo · 2024-11-03

**Soundness:** 2
**Presentation:** 3
**Contribution:** 3
**Rating:** 5
**Confidence:** 3

**Summary:**

The paper proposes confidence estimation in monocular depth estimation and completion. The authors introduce a new method termed harmonious convergence estimation to integrate confidence estimation and for regression tasks taking training consistency.

- Introducing a harmonious convergence estimation and intra-batch convergence for monocular depth estimation and completion.
- Conducting diverse experiments across multiple datasets

**Strengths:**

S1. Clear Contribution with Confidence Estimation. The paper presents a contribution by incorporating confidence estimation for monocular depth estimation and completion.

S2. Diverse Experiments: The paper shows diverse experiments with two different datasets

S3. Writing and Presentation: The paper is well-written and easy to understand.

**Weaknesses:**

The paper lacks experimental results showing depth performance using commonly employed metrics in the depth estimation and completion research fields, such as δ (delta) or MAE.
This paper claims that the proposed confidence estimation method works well even in the more challenging task of regression compared to classification.
Therefore, it is necessary to show more clearly that the performance of depth estimation or completion improves according to the confidence estimation even if this paper's main focus is related to confidence estimation.

**Questions:**

I would like to start by thanking the authors for their contribution to this field with their submission.

This relates to Weakness. Is there any reason that those metrics are skipped in the experimental results?

---

> ### Author Response · Authors · 2024-11-25
> **Reply to reviewer LDwo**
>
> We sincerely thank the reviewer LDwo for valuable feedback and for recognizing the strengths of our work, including the clear contribution of confidence estimation, the diversity of experiments, and the clarity of the writing and presentation.
>
> In this paper, we utilize SILog and Abs Rel metrics for depth estimation and RMSE for depth completion.
> These metrics are adopted as the main ranking metrics in KITTI Vision Benchmark Leaderboard[1][2].
> In response to the reviewer's feedback, we have also included experimental results using metrics such as δ (delta) and MAE in the revised version of the paper. The revised paper shows that SILog, Abs Rel and RMSE demonstrate consistency with the δ (delta) and MAE in evaluating depth estimation and completion performance, respectively.
>
> The main goal of this paper is to propose a harmonious convergence estimation approach for confidence estimation in regression tasks (depth estimation and completion) instead of aiming for improved regression accuracy.  Therefore, we prioritized representative metrics, such as Pearson, Spearman correlation coefficient and AUSE, which align with the scope of our contribution to evaluate the performance of depth estimation and completion. We believe these results further reinforce the validity of our proposed approach and demonstrate its impact on depth estimation and completion of tasks.
>
> [1] https://www.cvlibs.net/datasets/kitti/eval_depth.php?benchmark=depth_completion
>
> [2] https://www.cvlibs.net/datasets/kitti/eval_depth.php?benchmark=depth_prediction

---

> > ### Comment · Reviewer_LDwo · 2024-11-26
> >
> > I appreciate the authors' efforts in addressing the inclusion of additional metrics like delta and MAE.
> > However, I still find it unclear what the ultimate task that confidence estimation is aiming to address is.
> > If the primary objective of confidence estimation is to enhance the reliability of depth prediction or completion, then further validation for depth estimation or completion is necessary.
> > If not, the authors should have clarified the goal of confidence estimation.
> > Therefore, I regret that I cannot increase the score at this time.

---

> > > ### Author Response · Authors · 2024-11-26
> > >
> > > Thank you for your feedback and for raising this important point.
> > >
> > > The main objective of our paper is to address the issue of **overconfident predictions** [1][2][3] in depth prediction tasks. We understand that large errors in certain depth prediction regions are inevitable; however, we hope to assign low confidence to such regions, avoiding large errors with high confidence (overconfidence). Specifically, we aim to ensure that depth predictions labeled as "high confidence" correspond to more accurate predictions (analogous to true positives in classification), while depth predictions with higher errors are appropriately assigned "low confidence" (analogous to false positives) [3]. Therefore, the correlation between confidence estimation and depth prediction is our primary evaluation metric in this paper.
> > >
> > > Recent advances, such as training consistency (Li et al., 2023) [4] and prediction consistency (Moon et al., 2020) [5], show promise in mitigating overconfidence in classification tasks by leveraging training information through additional regularization. Extending this concept from classification to dense regression tasks such as depth estimation and completion presents two key challenges: spatial misalignment and consistency calculation.
> > >
> > > Our primary contribution lies in bridging the gap for introducing training consistency into regression tasks, which requires novel designs, such as intra-batch convergence estimation and harmonious convergence loss. Experimental results verify the effectiveness of our proposed method.
> > >
> > > [1] Matthias Hein, Maksym Andriushchenko, and Julian Bitterwolf. Why ReLU networks yield high-confidence predictions far away from the training data and how to mitigate the problem. In Proceedings of the IEEE Conference on Computer Vision and Pattern Recognition (CVPR), pages 41–50, 2019.
> > >
> > > [2] Meronen L, Trapp M, Pilzer A, et al. Fixing overconfidence in dynamic neural networks[C]//Proceedings of the IEEE/CVF Winter Conference on Applications of Computer Vision. 2024: 2680-2690.
> > >
> > > [3] Melotti G, Premebida C, Bird J J, et al. Reducing overconfidence predictions in autonomous driving perception[J]. IEEE Access, 2022, 10: 54805-54821.
> > >
> > > [4] Chen Li, Xiaoling Hu, and Chao Chen. Confidence estimation using unlabeled data. In The Eleventh International Conference on Learning Representations (ICLR), 2023.
> > >
> > > [5] Moon J, Kim J, Shin Y, et al. Confidence-aware learning for deep neural networks[C]//international conference on machine learning. PMLR, 2020: 7034-7044.

---

### Comment · Area_Chair_KZFw · 2024-11-27
**Reminder: Last day for author feedback**

This is a reminder that today is the last day allotted for author feedback. If there are any more last minute comments, please send them by today.

---

### Meta-Review · Area_Chair_KZFw · 2024-12-18

**Metareview:**

The authors proposed an confidence estimation (related to uncertainty quantification) approach for monocular depth completion and estimation. The main claim is that the proposed harmonious (intra-batch) convergence improves confidence of predictions. Reviewers T3bM and had4 brought up critical point regarding the motivation of why (theoretic) one would take this approach and what contributes to the gain, to which the author answered by re-describing the approach and stating the reason as being an extension of [1]; while T3bM and had4 did not reply, we do not think these responses adequately address the questions but bring up concerns that the approach may be incremental. Additionally, Reviewer LDwo brought up the need for additional evaluation metrics, we find that this is helpful to improve the paper but not critical. However, the comparisons made are of higher concern. Uncertainty quantification has been widely studied for both monocular depth completion and estimation. The choice of methods for comparisons were intended for broad range of tasks (with the exception of [F] which is for monocular depth estimation) and not necessarily depth completion or estimation. If the goal is to obtain better confidence or uncertainty then the experiments should at least include the tasks considered by the previous work. If the goal is to obtain better confidence or uncertainty specifically for depth completion and estimation then the authors should consider existing literature such as [A, B, C, D].

[A] Qu et al. Bayesian Deep Basis Fitting for Depth Completion with Uncertainty. ICCV 2021.

[B] Eldesokey et al. Uncertainty-aware cnns for depth completion: Uncertainty from beginning to end. CVPR 2020.

[C] Poggi et al. On the uncertainty of self-supervised monocular depth estimation. CVPR 2020.

[D] Marsal et al. MonoProb: Self-Supervised Monocular Depth Estimation With Interpretable Uncertainty. WACV 2024.

[E] Li et al. Confidence estimation using unlabeled data. ICLR 2023.

[F] Hornauer and Belagiannis. Gradient-based uncertainty for monocular depth estimation. ECCV 2022.

**Additional Comments On Reviewer Discussion:**

Reviewer LDwo raised concerns regarding evaluation for depth completion and estimation performance. While this is useful for improving the manuscript, we did not find this to be critical. Reviewers T3bM and had4 asked for clarification as to why the authors took this approach and what contributes to the gain reported. The authors responded by re-describing the approach and stating the reason as being an extension of [E]; while T3bM and had4 did not reply, we do not think these responses address the questions but bring up concerns that the approach may be incremental. T3bM also raised concerns regarding existing literature, while had4 had a minor concern regarding Fig. 1.

[E] Li et al. Confidence estimation using unlabeled data. ICLR 2023.

---

### Decision · Program_Chairs · 2025-01-22

Reject